A multi-class classification model for supporting the diagnosis of type II diabetes mellitus

http://orcid.org/0000-0002-8552-3016 Kuo Kuang-Ming 1
Talley Paul 2
Kao YuHsi 3
Huang Chi Hsien 4 5 evaairgigaa@gmail.com
1 Department of Healthcare Administration, I-Shou University , Kaohsiung City , Taiwan, Republic of China
2 Department of Applied English, I-Shou University , Kaohsiung City, Taiwan , Republic of China
3 Department of Endocrinology, E-Da Hospital , Kaohsiung City, Taiwan , Republic of China
4 Department of Family Medicine, E-Da Hospital, I-Shou University , Kaohsiung City, Taiwan , Republic of China
5 Department of Community Healthcare and Geriatrics, Nagoya University Graduate School of Medicine , Nagoya , Japan
Palazón-Bru Antonio
Electronic publication date: 2020 Sep 10
Publication date: 2020
Volume: 8
Electronic Location ID: e9920
Received 2020 May 21; Accepted 2020 Aug 20
Copyright: © 2020 Kuo et al.
Copyright year: 2020
Copyright holder: Kuo et al.
License: This is an open access article distributed under the terms of the Creative Commons Attribution License, which permits unrestricted use, distribution, reproduction and adaptation in any medium and for any purpose provided that it is properly attributed. For attribution, the original author(s), title, publication source (PeerJ) and either DOI or URL of the article must be cited.
License URL: https://creativecommons.org/licenses/by/4.0/

Keywords: Diagnosis, Machine-learning techniques, Predictive models, Type 2 diabetes mellitus

Funding: E-Da Hospital, Kaohsiung, Taiwan ISU-106-IUC-05 This study was supported by E-Da Hospital, Kaohsiung, Taiwan under grant number ISU-106-IUC-05. The funders had no role in study design, data collection and analysis, decision to publish, or preparation of the manuscript.

==============================
Background

Numerous studies have utilized machine-learning techniques to predict the early onset of type 2 diabetes mellitus. However, fewer studies have been conducted to predict an appropriate diagnosis code for the type 2 diabetes mellitus condition. Further, ensemble techniques such as bagging and boosting have likewise been utilized to an even lesser extent. The present study aims to identify appropriate diagnosis codes for type 2 diabetes mellitus patients by means of building a multi-class prediction model which is both parsimonious and possessing minimum features. In addition, the importance of features for predicting diagnose code is provided.

Methods

This study included 149 patients who have contracted type 2 diabetes mellitus. The sample was collected from a large hospital in Taiwan from November, 2017 to May, 2018. Machine learning algorithms including instance-based, decision trees, deep neural network, and ensemble algorithms were all used to build the predictive models utilized in this study. Average accuracy, area under receiver operating characteristic curve, Matthew correlation coefficient, macro-precision, recall, weighted average of precision and recall, and model process time were subsequently used to assess the performance of the built models. Information gain and gain ratio were used in order to demonstrate feature importance.

Results

The results showed that most algorithms, except for deep neural network, performed well in terms of all performance indices regardless of either the training or testing dataset that were used. Ten features and their importance to determine the diagnosis code of type 2 diabetes mellitus were identified. Our proposed predictive model can be further developed into a clinical diagnosis support system or integrated into existing healthcare information systems. Both methods of application can effectively support physicians whenever they are diagnosing type 2 diabetes mellitus patients in order to foster better patient-care planning.

Introduction

Diabetes mellitus (DM), as defined by the American Diabetes Association (2010), refers to a group of metabolic disorders primarily induced by impaired insulin secretion and/or action. Insulin deficiency and increased insulin resistance may lead to an elevated blood glucose level and impaired metabolism of carbohydrates, fat, and protein (American Diabetes Association, 2010). DM is one of the most prevalent endocrine disorders, influencing more than 200 million people universally (Kavakiotis et al., 2017). DM has therefore become a global public health challenge, and it is a key health concern worldwide. DM is expected to increase dramatically, and it could potentially be the seventh-leading reason of death in 2030 (World Health Organization, 2016). In terms of health-related issues, DM can lead to other serious medical complications such as chronic kidney disease, acute kidney injury, cardiovascular disease, ischemic heart disease, stroke or even to death (World Health Organization, 2016). The direct and indirect estimated total cost of diabetes management in the U.S. in 2012 was $245 billion and increased to $327 billion in 2017 (Centers for Diseases Control & Prevention, 2017). The burden of DM is rapidly increasing on a global basis and has become a major public health concern. On the other hand, despite the possibly-related complications, DM can be appropriately managed with a comprehensive care plan, such as with a reasonable lifestyle change and significant medication control (American Diabetes Association, 2015).

There are two prevalent types of DM, including type 1 diabetes and type 2 diabetes (T2DM), according to the etio-pathology of the disorder (Maniruzzaman et al., 2017). T2DM, accounting for 90% of DM patients, is the most common form of diabetes (Maniruzzaman et al., 2017). Several risk factors which include smoking, overweight and obesity, physical inactivity, high blood pressure, high cholesterol, and high blood glucose levels were reported to be associated with T2DM (Centers for Diseases Control & Prevention, 2017). However, the links between T2DM and some risk factors still remain unclear (Eckel et al., 2011). Currently, the diagnosis of T2DM can be based on elevated Hemoglobin A1c, high fasting or random plasma glucose, and a clinical manifestation of increased urinary frequency (polyuria), thirst (polydipsia), and hunger (polyphagia) (American Diabetes Association, 2010). However, it has been estimated that nearly 7.2 million people (23.8% of diabetes patients) remain undiagnosed in the United States (Centers for Diseases Control & Prevention, 2017). Hence, there is a rising need for related research to early identify and to confirm T2DM diagnosis more efficiently and accurately in clinical settings (Kagawa et al., 2017).

Information technologies such as machine-learning techniques have become a vital instrument in determining T2DM diagnosis and affecting management for health care providers and patients (Rigla et al., 2017). Numerous studies have utilized machine-learning techniques to predict the onset of T2DM. While previous DM prediction studies have shown a potential for detecting the onset of T2DM (Alghamdi et al., 2017; Anderson et al., 2015; Esteban et al., 2017; Kagawa et al., 2017; Maniruzzaman et al., 2017; Nilashi et al., 2017; Pei et al., 2019; Talaei-Khoei & Wilson, 2018; Upadhyaya et al., 2017; Wu et al., 2018), no studies, to our knowledge, have been aimed at predicting a suitable diagnosis code for T2DM patients. Further, ensemble machine-learning techniques such as bagging and boosting approaches are less utilized in these studies (Esteban et al., 2017). Most importantly, less multi-class studies, to our knowledge, have been conducted (Esteban et al., 2017). Therefore, the intended purpose of this study is to leverage routinely available clinical data in order to establish a multi-class predictive model based on bagging and boosting machine-learning techniques useful to identify Asian T2DM patients with a corresponding diagnosis code. The major contribution of our proposed predictive model is its ability to identify a corresponding ICD-10-CM code, not just to identify the onset of T2DM. The correct identification of ICD-10-CM code for T2DM can help physicians and patients form a proper patient-care plan, thus improving the conditions of T2DM patients while reducing the associated heavy financial burden caused by T2DM.

The remainder of this article is organized as follows: In section 2, we briefly introduce artificial neural networks, decision trees, ensemble models, and support vector machine. In section 3, we present the review of T2DM related studies that used machine-learning techniques. In section 4, we explain the methodology used for data collection, preparation, and analysis in this study. In section 5, we present the results and in section 6, we discuss the findings of this study. Finally, in section 7, we summarize and conclude this study.

Machine Learning Algorithms

Artificial neural network

An artificial neural network (ANN) involves the development of models that enable computers to learn in ways similar to the human brain (Ciaburro & Venkateswaran, 2017; Larrañaga et al., 2019). An ANN is usually organized in layers which comprise a number of interconnected and weighted nodes (or neurons) (Clark, 2013; Lantz, 2015). To constitute an ANN, at least three layers, including an input layer, a hidden layer, and an output layer, should be included (Lewis, 2016).

Figure 1 shows the relations between input nodes (xi) and the output node (y). Each of the input nodes is weighted (wi) based on its importance (Beysolow, 2017). The input nodes are then summed and passed on according to the activation function (Clark, 2013; Lantz, 2015). An activation is the mechanism by which the artificial neuron handles incoming information and disseminates it all over the network (Lewis, 2016).

Figure 1 Artificial neural network architecture.

Decision trees

Decision trees utilize a tree structure to model the associations found among features and the possible outcomes (Provost & Fawcett, 2013). As Fig. 2 shows, a decision to be considered starts at the root node (Faul, 2020), and a decision is made based on the questions of whether the value is higher or lower than a threshold (Brownlee, 2017). These decisions then split the data across branches indicating likely outcomes of a decision (Clark, 2013). If a final decision can be reached, the tree is terminated by terminal nodes (Faul, 2020). There are many implementations of decision trees, one of the most famous is the C5.0 algorithm, an improvement of C4.5 algorithm (Quinlan, 1996), and has become a de-facto standard to create decision trees.

Figure 2 Decision trees.

Ensemble model (bagging and boosting)

The technique of merging and managing the predictions of multiple models is known as an ensemble approach (Lantz, 2015). More specifically, ensemble methods are hinged on the notion that by merging multiple weaker learners, a stronger learner is generated (Clark, 2013). Bagging and boosting are widespread accepted ensemble methods currently.

Bagging

One of the ensemble approaches to receive widely acknowledgement adopted a technique named bootstrap aggregating or bagging, to generate a number of datasets for training by bootstrap sampling from the primitive training dataset (Lantz, 2015). These data are then utilized to create a set of models with each incorporating only one classifier. Averaging (for numeric prediction) or voting (for classification) are used to determine the model’s terminal predictions (Beysolow, 2017; Clark, 2013). Among many bagging classifiers, random forest, a combination of several decision trees (Beysolow, 2017), merges the basic rules of bagging with random feature selection to increase additional variety to the building of the models. After the ensembles of trees is created, the model utilizes a vote to merge the tree’s predictions (Beysolow, 2017).

Boosting

Another ensemble-based method is known as boosting since it is a method of boosting weak learners to become strong learners (Beysolow, 2017; Brownlee, 2017; Faul, 2020). In boosting, each new tree is a fit on an adjusted version of the primitive dataset. Different from bagging, boosting resampled datasets are constructed to generate complementary learners, and boosting gives each learner’s vote a weight based on its past performance (Lantz, 2015).

Among the many boosting classifiers, eXtreme gradient boosting (Chen & Guestrin, 2016) is one of the most popular applications of gradient boosting concept. This classifier is basically designed to enhance the performance and speed of a machine learning model. What makes eXtreme gradient boosting peculiar is that it utilizes a more regularized model formalization to regulate over-fitting, which thus gives it better performance (Lantz, 2015).

Support vector machine

A support vector machine (SVM), an instance-based algorithm, tries to maximize the margin between two classes by using kernel function (Marsland, 2015). In other words, SVM creates a boundary called a hyperplane (Beysolow, 2017) and tries to search for the maximum margin hyperplane (Brownlee, 2017), which breaks the space to create the best homogenous partitions on two different classes (see Fig. 3). The support vectors are the points from each class that are the nearest to the maximum margin hyperplane, which is a key feature of SVMs (Lantz, 2015). SVMs can be utilized along with almost any type of learning task, including numeric prediction and classification (Kuhn & Johnson, 2013).

Figure 3 Margin hyperplane, support vector, and convex hull.

Related Work

Thus far, a large number of studies have attempted to predict the onset of T2DM based on differing machine-learning algorithms. The study by Kavakiotis et al. (2017) provides an excellent review on machine-learning and data-mining methods in prior diabetes research. To prevent reiteration of what Kavakiotis et al. (2017) have found, we reviewed T2DM studies made during 2015–2019 that utilized machine-learning techniques, as follows: (1) adopted machine-learning algorithms, (2) the features used to predict T2DM, (3) the sample locations experienced, and (4) classification type.

Adopted machine learning algorithms

Several types of machine-learning algorithms, including instance-based (Esteban et al., 2017; Kagawa et al., 2017; Nilashi et al., 2017; Pei et al., 2019; Talaei-Khoei & Wilson, 2018), decision trees (Alghamdi et al., 2017; Esteban et al., 2017; Pei et al., 2019; Talaei-Khoei & Wilson, 2018), artificial neural network (Esteban et al., 2017; Nilashi et al., 2017; Talaei-Khoei & Wilson, 2018), ensemble (Alghamdi et al., 2017; Esteban et al., 2017; Pei et al., 2019), Bayesian (Alghamdi et al., 2017; Anderson et al., 2015; Esteban et al., 2017; Maniruzzaman et al., 2017; Pei et al., 2019), statistical model (Alghamdi et al., 2017; Esteban et al., 2017; Maniruzzaman et al., 2017; Talaei-Khoei & Wilson, 2018; Wu et al., 2018), and others (see Table 1), have been adopted to predict T2DM-related issues. However, these studies revealed different results in predicting the onset of T2DM even with the same machine-learning algorithm. For example, Pei et al. (2019) and Alghamdi et al. (2017) both adopted J48 as one of their algorithms for predicting the onset of T2DM, only Pei et al. (2019) found that J48 had the best performance. The performance of support vector machine also differs among opposing studies (Esteban et al., 2017; Kagawa et al., 2017; Nilashi et al., 2017; Pei et al., 2019; Talaei-Khoei & Wilson, 2018). Further, not all inclusionary studies adopted the same algorithms, making it difficult to accurately compare the performance of differing algorithms. Finally, artificial neural network were adopted by three studies (Esteban et al., 2017; Nilashi et al., 2017; Talaei-Khoei & Wilson, 2018) and outperformed other algorithms in these respective studies. Further, deep-learning techniques were not applied in these T2DM-related studies.

Table 1 Type 2 diabetes mellitus diagnosis related studies: adopted machine learning algorithms.

Study	Instance-based	Decision trees	Neural network	Ensemble	Bayesian	Statistical model	Others	
Pei et al. (2019)	Support vector machine	J48*		Adaboostm1	Naïve Bayes, Bayes net			
Wu et al. (2018)						Logistic regression	K-means	
Talaei-Khoei & Wilson (2018)	Support vector machine*	Decision
trees	Neural network*			Logistic regression*	Clustering	
Upadhyaya et al. (2017)							First-order logic rules	
Nilashi et al. (2017)	Self-organizing map, support vector machine		Neural network*				Principal component analysis	
Maniruzzaman et al. (2017)					Naïve Bayes	Linear discriminant analysis, Quadratic discriminant analysis	Gaussian process classification*	
Kagawa et al. (2017)	Support vector machine						Rule-based*, Modified PheKB	
Alghamdi et al. (2017)		J48, Decision tree, Logistic model tree		Random forest	Naïve Bayes	Logistic regression*		
Esteban et al. (2017)	Support vector machine, KNN	C5.0	Neural networks*	Random forest, Gradient boosting machine, Extreme gradient boosting	Bayesian model	Linear model, Discriminant analysis, Partial least squares, Multinomial logistic regression	Rule-based, Elastic net, Nearest shrunken centroid	
Anderson et al. (2015)					Bayesian inference			
Note:

* Denotes the best performed algorithm.

Features used to predict T2DM

Regarding features selected to predict the onset of T2DM, they can be roughly classified into five major categories (see Table 2): (1) demographic data; (2) laboratory test results; (3) vital signs; (4) life style; and, (5) history. Demographic data such as age, gender, Body Mass Index (BMI) were often adopted features used for predicting the onset of T2DM (Alghamdi et al., 2017; Anderson et al., 2015; Pei et al., 2019; Talaei-Khoei & Wilson, 2018). Laboratory tests such as fast plasma glucose, Hemoglobin A1c, high-density lipoprotein cholesterol were commonly seen features used by the regarded T2DM studies (Anderson et al., 2015; Maniruzzaman et al., 2017; Talaei-Khoei & Wilson, 2018; Upadhyaya et al., 2017; Wu et al., 2018). Vital signs such as diastolic blood pressure and systolic blood pressure were also used by those reviewed studies (Alghamdi et al., 2017; Nilashi et al., 2017; Talaei-Khoei & Wilson, 2018; Wu et al., 2018). Life-style features such as physical activity, work stress, salty-food preference (Pei et al., 2019), shortness of breath, frequent urination at night, excessive thirst (Maniruzzaman et al., 2017; Nilashi et al., 2017; Talaei-Khoei & Wilson, 2018), and sedentary lifestyle (Alghamdi et al., 2017) were also included. Finally, history features such as family history of diabetes (Maniruzzaman et al., 2017; Nilashi et al., 2017; Pei et al., 2019; Talaei-Khoei & Wilson, 2018) or prescriptions of diabetes-related medication (Esteban et al., 2017; Kagawa et al., 2017; Upadhyaya et al., 2017) were included. In practice, it is not easy to collect all the features utilized in the above-discussed studies since some features require extra efforts to acquire. It will therefore be more practical and efficient to collect required data, for prediction of the onset of T2DM, from Electronic Medical Records since most hospitals have extensively used Electronic Medical Records to assist patient care.

Table 2 Type 2 diabetes mellitus diagnosis-related studies: Included features for machine learning.

Study	Demographic data	Laboratory test results	Vital signs	Life style	History	Others	
Pei et al. (2019)	Age, gender, BMI			Physical activity, work stress, salty food preference	History of cardiovascular disease or stroke, family history of diabetes, hypertension		
Wu et al. (2018)	Age, BMI	2-h plasma glucose, 2-h
serum insulin, diabetes pedigree function	Diastolic
blood
pressure		Number of times pregnant	Triceps skin fold thickness	
Talaei-Khoei & Wilson (2018)	Age, sex, BMI	High density lipoprotein cholesterol, triglycerides,
fast plasma glucose, Hemoglobin A1c	Systolic
blood
pressure	Shortness of breath, frequent urination at night, excessive thirst	History of high blood glucose, parental history of diabetes	Waist/hip ratio, waist circumference	
Upadhyaya et al. (2017)		Hemoglobin A1c			A prescription for metformin, DM-related medication	ICD-9-CM code	
Nilashi et al. (2017)	Age, sex, BMI	High density lipoprotein cholesterol, triglycerides,
fast plasma glucose, Hemoglobin A1c	Systolic
blood
pressure	Shortness of breath, frequent urination at night, excessive thirst	History of high blood glucose, parental history of diabetes	Waist/hip ratio, waist circumference	
Maniruzzaman et al. (2017)	Age, sex, BMI	High density lipoprotein cholesterol, triglycerides, fast plasma glucose, Hemoglobin A1c	Systolic
blood
pressure	Shortness of breath, frequent urination at night, excessive thirst	History of high blood glucose, parental history of diabetes	Waist/hip ratio, waist circumference	
Kagawa et al. (2017)		Random glucose, glycol-albumin, HbA1c, GAD antibody, IA2 antibody, C-peptide			T2DM medication	ICD-10 code, I25l-insulin biding ratio	
Alghamdi et al. (2017)	Age, black, obesity	Metabolic equivalent	Resting heart rate, resting systolic blood pressure, resting diastolic blood pressure	Sedentary lifestyle	Family history of premature coronary artery disease, hypertension, aspirin		
Esteban et al. (2017)		Fasting glycemia, HbA1c			Diabetes Mellitus related prescriptions filled	Diabetes Mellitus related codes	
Anderson et al. (2015)	Age, gender, BMI, race, region, insurance status, average annual household income, education status	Hemoglobin A1c, fasting glucose, 2h oral glucose tolerance, random glucose, triglycerides, total bilirubin, alanine aminotransferase, creatinine, low-density lipoprotein, high-density lipoprotein	Heart rate, blood pressure, body temperature				

Sample locations experienced

As shown in Table 3, we can find that most samples were taken in the United States (Alghamdi et al., 2017; Anderson et al., 2015; Maniruzzaman et al., 2017; Nilashi et al., 2017; Upadhyaya et al., 2017; Wu et al., 2018), while fewer study samples were drawn from Asian countries (Kagawa et al., 2017; Pei et al., 2019). The evidence reviewed here clearly highlights the present need that more diversified samples should be included and analyzed aiming to better clarify their relations with the onset of T2DM.

Table 3 Type 2 diabetes mellitus diagnosis-related studies: samples and classification type.

Study	Country	Sample size	Classification type	Results	
Pei et al. (2019)	China	4,205	Binary	J48 has the best performance (accuracy = 0.9503, precision = 0.950, recall = 0.950, F-measure = 0.948, and AUC = 0.964)	
Wu et al. (2018)	USA	768	Binary	The proposed model attained a 3.04% higher prediction accuracy than those of other studies	
Talaei-Khoei & Wilson (2018)	Australia	10,911	Binary	The performance of different learners depends on both period and purpose of prediction	
Upadhyaya et al. (2017)	USA	4,208	Binary	The proposed algorithm performed well with a 99.70% sensitivity and a 99.97% specificity	
Nilashi et al. (2017)	USA	768	Binary	The proposed method remarkably improves the accuracy of prediction in relation to prior methods	
Maniruzzaman et al. (2017)	USA	768	Binary	The performance of Gaussian process classification are better than other methods with accuracy = 81.97%, sensitivity = 91.79%, positive predictive value = 84.91%, and negative predictive value = 62.50%	
Kagawa et al. (2017)	Japan	104,522	Binary	The proposed phenotyping algorithms show better performance than baseline algorithms	
Alghamdi et al. (2017)	USA	32,555	Binary	The proposed ensemble approach achieved high accuracy of prediction (AUC = 0.920)	
Esteban et al. (2017)	Argentina	2,463	Multi-class	The stacked generalization strategy and feed-forward neural network performed the best with validation set	
Anderson et al. (2015)	USA	24,331	Binary	The proposed ensemble model accurately predicted progression to T2DM (AUC = 0.76), and was validated out of sample (AUC = 0.78)	

Classification type

Finally, most of the reviewed studies predicted only two classes—the onset of T2DM or not (see Table 3), while the multi-class application was seen less often (Esteban et al., 2017). In clinical practice, it is apparently insufficient to diagnose T2DM with only two classes, so a multi-class classification model is therefore required to diagnose the differing types of T2DM in a more accurate manner in order to provide personalized patient care coupled with precision medicine.

According to a review of recent T2DM studies that utilized machine-learning techniques, several points should be duly noted. First, how different algorithms perform in predicting the onset of T2DM is still unclear and incomparable since each of the studies adopted differing algorithms. Second, deep learning and ensemble approaches are utilized to a lesser extent than in those reviewed studies. Third, no clear results demonstrate which features should be used to predict the onset of T2DM. Fourth, T2DM patients from Asian countries were under-represented in studies using machine-learning techniques than from outside the U.S. As a result, these findings are not comparable or contrastive in achieving a better understanding of the various aspects regarding T2DM. Fifth, owing to an increasing number of disease sub-categories, it is mandatory to conduct a multi-class study to facilitate and then to confirm final diagnosis. For example, most reviewed studies demonstrated the potential for predicting the onset of T2DM, whereas the onset of T2DM complications, including as retinopathy, neuropathy, and nephropathy, were rarely if ever investigated.

Materials and Methods

Data

Diagnosing T2DM depends primarily on laboratory test results (American Diabetes Association, 2010), we therefore required a collection of those data from T2DM patients. A plausible T2DM patient list was first obtained, containing patients who had visited an endocrinologist (one of our authors) between November, 2017 and May, 2018 at a large hospital in southern Taiwan. The Institutional Review Board of E-Da Hospital approved our study protocol and waived informed consent regarding this study (EMRP-107-048). In consideration of the features to be included, we elected to adopt 10 common features based on our review of prior studies related to DM prediction models (Anderson et al., 2015; Pei et al., 2019; Talaei-Khoei & Wilson, 2018; Wu et al., 2018). These readily available features can be drawn directly or indirectly from the content of Electronic Medical Records. By doing so, the predictive model we proposed can be adopted by most hospitals since these selected features are already stored in existing databases.

The 10 health-related features can be primarily classified into two categories: demographic data and laboratory test results. Demographic data included age, gender, smoking status, and BMI which were reported to be associated with the onset of T2DM (Yuan et al., 2018). On the other hand, laboratory data are comprised of total cholesterol, triglyceride, glucose (AC), Hemoglobin A1c, high-density lipoprotein cholesterol, and low-density lipoprotein cholesterol which were indicators of impaired metabolic function pre-disposing DM (Guasch-Ferré et al., 2016).

Eligibility criteria for the study were that a patient must (1) be diagnosed through an international classification of diseases, tenth revision, clinical modification (ICD-10-CM) starting with E11, and (2) no missing data in total cholesterol, triglyceride, glucose (AC), Hemoglobin A1c, high-density lipoprotein cholesterol, and low-density lipoprotein cholesterol was evident. Initially, a total of 10,527 plausible T2DM patient information were obtained and duplicated patient listings were first removed. Patients with missing laboratory test results were then removed. Since there may be many ICD-10-CM codes utilized for diagnosing T2DM, we limited our predicted classes to the first five digits of the ICD-10-CM code; and as such, these five-digit codes must be among the top ICD-10-CM codes appearing in our collected data. Finally, 149 eligible records, including E1121 (T2DM with diabetic nephropathy, n = 45), E1143 (T2DM with diabetic autonomic [poly]neuropathy, n = 88), and E1165 (T2DM with hyperglycemia, n = 16), without missing values were collected.

Our inclusion of these 10 features primarily differs from prior T2DM-related studies in that we only included demographic data and laboratory test results, while prior T2DM studies included a wider variety of data. In words, we aimed to build a parsimonious predictive model possessing minimum features. Table 4 shows the detailed operational definition of features used in our study.

Table 4 Operational definition of features.

Features/Target class	Measurement	Definition	References	
Target class	Diagnosis of T2DM	Discrete	The probability of four kinds of T2DM diagnosis: E1121, E1143, and E1165	NA	
Features	Gender	Discrete	Gender of the patients, Male or Female.	Anderson et al. (2015), Pei et al. (2019), Wu et al. (2018)	
Age	Continuous	Age (in years) during out-patient services	Anderson et al. (2015), Pei et al. (2019), Talaei-Khoei & Wilson (2018)	
Smoking status	Discrete	Yes, quit, or no		
BMI	Continuous	Body mass index	Anderson et al. (2015), Pei et al. (2019), Wu et al. (2018)	
Total Cholesterol	Continuous	The level of total cholesterol during out-patient services		
Triglyceride	Continuous	The level of triglyceride during out-patient services	Anderson et al. (2015), Talaei-Khoei & Wilson (2018)	
	Glucose (AC)	Continuous	The level of glucose (AC) during out-patient services	Anderson et al. (2015)	
Hemoglobin A1c	Continuous	The level of Hemoglobin A1c during out-patient services	Anderson et al. (2015), Kagawa et al. (2017), Talaei-Khoei & Wilson (2018)	
High density lipoprotein cholesterol	Continuous	The level of high-density lipoprotein cholesterol during out-patient services	Anderson et al. (2015), Talaei-Khoei & Wilson (2018)	
Low density lipoprotein cholesterol	Continuous	The level of low-density lipoprotein cholesterol during out-patient services	Anderson et al. (2015)	

Experimental setup

To predict a diagnosis code for the T2DM patient, we adopted R 4.0.0 software (R Core Team, 2020) for purposes of data analysis. Since our data is non-linear, machine-learning techniques are well-suited for predicting the ICD-10-CM code of T2DM. Based on the methodological gaps found in our review of T2DM related studies, we decided to choose five machine-learning algorithms including instance-based (Support vector machine), decision trees (C5.0), deep neural network, and ensemble (Random forest and eXtreme gradient boosting) as primary learners in our study.

We used the mlr 2.17.1 package (Bischl et al., 2016) to automatically tune the optimal model parameters for these four learners aiming to obtain a better level of predictive performance. The R packages used for machine-learning algorithms and their respective optimal model parameters are shown in Table 5. Further, since our predicted class is imbalanced, we utilized a synthetic minority over-sampling technique provided by UBL package (Branco, Ribeiro & Torgo, 2016) in order to improve the model performance.

Table 5 R packages used and the optimal model parameters given.

Method	Parameters	Best parameter setting	R packages	
Support vector machine	sigma	0.664667494	kernlab 0.9-29	
C	11.07262251		
C5.0	winnow	FALSE	C50 0.1.3	
trials	43		
Deep neural network	hidden	200	h2o 3.30.0.1	
input_dropout_ratio	0		
activation	Maxout		
eXtreme gradient boosting	nrounds	154	xgboost 1.0.0.2	
max_depth	10		
eta	0.745922343		
gamma	3.194824195		
colsample_bytree	0.945590117		
min_child_weight	3.35705624		
subsample	0.802348509		
Random Forest	mtry	2	randomForest 4.6-14	

We adopted: (1) 10-fold cross-validation; (2) leave-one-subject-out; and (3) holdout approaches to assess the performance of the five learners. The 10-fold cross-validation approach randomly splits the dataset into 10 subsets with roughly similar sizes, among which nine subsets are used for constructing the model and the remaining one subset is utilized for testing the model (Provost & Fawcett, 2013). Leave-one-subject-out cross-validation is a special case of k-fold cross-validation since k is the number of samples while holdout simply splits data into training samples for building the predictive model and testing samples for estimating model performance (Kuhn & Johnson, 2013).

Performance metrics

To better evaluate the performance of a multi-class setting, we employed average accuracy, area under receiver operating characteristic (AUC), Matthew correlation coefficient (MCC), and the macro-averaging of precision, recall, and F1 score (weighted average of precision and recall) according to the suggestions taken from the literature (Sokolova & Lapalme, 2009). These metrics were measured based on a confusion matrix (see Table 6).

Table 6 Confusion matrix.

		Predicted class	
	Positive	Negative	
Actual class	Positive	True positive (TP)	False negative (FN)	
Negative	False positive (FP)	True negative (TN)	

The average accuracy, MCC, micro- and macro-averaging precision, recall, and F1 score were then acquired using the formulae located in Table 7.

Table 7 Formulae for performance metrics.

Metric	Formula	
Average accuracy	∑i=1lTPi+TNiTPi+FNi+FPi+TNil	
Matthew correlation coefficient	(TP∗TN+FP∗FN)Sqrt((TP+FP)∗(TP+FN)∗(TN+FP)∗(TN+FN))	
PrecisionM	∑i=1lTPi(TPi+FPi)l	
RecallM	∑i=1lTPi(TPi+FNi)l	
F1 scoreM	2∗PrecisionM∗RecallMPrecisionM+RecallM	
Note:

l denotes class levels, M denotes macro-averaging metrics, TP means true positive, FP denotes false positive, FN means false negative, and TN denotes true negative.

Regarding the interpretation of these metrics, the average accuracy, AUC, MCC, macro-averaging and micro-averaging precision, recall, and F1 score value between 0 and 1, with values approaching 1, imply better performance.

Results

Data profiles

Table 8 demonstrates the descriptive statistics for T2DM patients. Among these figures, the proportion of the male sample is higher than that of female, aged 21–91 years, and most samples did not smoke, or had quit smoking, at the time of survey administration. Furthermore, the average BMI of samples belonging to the “obesity” level, and the average levels of glucose (AC) and Hemoglobin A1c are higher than the normal values. On average, other laboratory test results fall inside the normal range.

Table 8 Data summary results.

Feature	Range	Summary statistics	
Gender	Male/Female	Male: 86, Female: 63	
Age	21~91	M = 61.27, SD = 13.70	
Smoking status	No/Quit/Yes	No = 123, Quit = 10, Yes = 16	
BMI	15.49~44.05	M = 26.63, SD = 4.77	
Total cholesterol	77~311	M = 151.98, SD = 34.38	
Triglyceride	37~546	M = 136.64, SD = 93.30	
Glucose (AC)	68~346	M = 146.58, SD = 51.72	
Hemoglobin A1c	5.1~11.6	M = 7.46, SD = 1.21	
High density lipoprotein cholesterol	16~98	M = 47.44, SD = 14.87	
Low density lipoprotein cholesterol	29~152	M = 71.42, SD = 25.66	
Note:

M denotes mean and SD means standard deviation.

Model performance

Under 10-fold cross-validation, the performance of support vector machine ranked the highest in accuracy, AUC, MCC, macro-averaging F1 score, macro-averaging precision, and macro-averaging recall metrics with training samples (see Table 9). This was followed by random forest, C5.0, deep neural network, and eXtreme gradient boosting. Further, the process time for training the support vector machine was also the shortest compared to the remaining algorithms. When comparing the performance of the five trained models in the test samples, support vector machine, C5.0, and random forest perfectly achieved one in accuracy, AUC, MCC, macro-averaging F1 score, macro-averaging precision, and macro-averaging recall metrics (see Table 9; Fig. 4). eXtreme gradient boosting learner also achieved higher than 0.9 in all metrics. Deep neural network however performed poorer than the other four learners in all metrics. We then compared the model performance by use of the Stuart–Maxwell test which is better suited for multi-class classification models than McNemar test (Maxwell, 1970; McNemar, 1947; Stuart, 1955). Since support vector machine, C5.0, and random forest perfectly predicted ICD-10-CM codes used for T2DM, we only statistically compared the performance of deep neural network and eXtreme gradient boosting learners. The Stuart–Maxwell tests demonstrated significant results for both deep neural network (p < 0.001) and eXtreme gradient boosting (p = 0.002), thus indicating significant difference disagreement between these two algorithms and the observed data.

Table 9 Model performance: 10-fold cross-validation.

Sample	Learner	Accuracy (SD)	AUC (SD)	MCC (SD)	Macro	Process time	Stuart–Maxwell test	
F1 (SD)	Precision (SD)	Recall (SD)	
Train	SVM	0.998 (0.006)	1.000 (0.000)	0.995 (0.011)	0.994 (0.012)	0.997 (0.008)	0.991 (0.015)	2.22		
C5.0	0.984 (0.015)	0.999 (0.001)	0.969 (0.031)	0.981 (0.020)	0.987 (0.015)	0.975 (0.026)	6.74		
DNN	0.947 (0.019)	0.985 (0.016)	0.896 (0.033)	0.935 (0.027)	0.956 (0.031)	0.922 (0.028)	13.56		
XGB	0.943 (0.021)	0.992 (0.008)	0.885 (0.044)	0.918 (0.050)	0.946 (0.036)	0.894 (0.058)	7.86		
RF	0.986 (0.010)	1.000 (0.000)	0.972 (0.017)	0.985 (0.011)	0.992 (0.006)	0.978 (0.016)	4.59		
Test	SVM	1.000	1.000	1.000	1.000	1.000	1.000			
C5.0	1.000	1.000	1.000	1.000	1.000	1.000			
DNN	0.855	0.985	0.730	0.678	0.876	0.684		χ2(3) = 253.20, p < 0.001	
XGB	0.989	1.000	0.979	0.985	0.992	0.978		χ2(2) = 13.00, p = 0.002	
RF	1.000	1.000	1.000	1.000	1.000	1.000			
Note:

AUC, area under receiver operating characteristic; SD, standard deviation; MCC, Matthew correlation coefficient; SVM, support vector machine; DNN, deep neural network; XGB, eXtreme gradient boosting; RF, random forest, the second is used to measure process time.

Figure 4 Model performance of test dataset—10-fold cross-validation.

AUC, area under receiver operating characteristic curve; MCC, Matthew correlation coefficient.

Under leave-one-subject-out cross-validation, both support vector machine and random forest performed better than the remaining classifiers, with training samples, in terms of all metrics, including process time (see Table 10). As for the model performance of testing samples, support vector machine, C5.0, and random forest perfectly achieved one in accuracy, AUC, MCC, macro-averaging F1 score, macro-averaging precision, and macro-averaging recall metrics (see Table 10; Fig. 5). Deep neural network and eXtreme gradient boosting still did not perform as well as the remaining classifiers. Stuart–Maxwell tests were then conducted for deep neural network and eXtreme gradient boosting. And, the results revealed that deep neural network still showed significant difference with the observed data (p < 0.001) while eXtreme gradient boosting showed insignificant difference with the observed data (p = 0.06).

Table 10 Model performance: leave-one-subject-out cross-validation.

Sample	Learner	Accuracy (SD)	AUC (SD)	MCC (SD)	Macro	Process time	Stuart–Maxwell test	
F1 (SD)	Precision (SD)	Recall (SD)	
Train	SVM	0.999 (0.000)	1.000 (0.000)	0.999 (0.000)	0.999 (0.000)	0.999 (0.000)	0.999 (0.000)	280.67		
C5.0	0.999 (0.000)	0.999 (0.000)	0.999 (0.000)	0.999 (0.000)	0.999 (0.000)	0.999 (0.000)	879.37		
DNN	0.984 (0.004)	0.998 (0.001)	0.968 (0.008)	0.981 (0.005)	0.983 (0.004)	0.979 (0.005)	2145.94		
XGB	0.992 (0.002)	0.999 (0.000)	0.985 (0.005)	0.990 (0.004)	0.994 (0.003)	0.986 (0.005)	1028.34		
RF	0.999 (0.000)	1.000 (0.000)	0.999 (0.000)	0.999 (0.000)	0.999 (0.000)	0.999 (0.000)	639.22		
Test	SVM	1.000	1.000	1.000	1.000	1.000	1.000			
C5.0	1.000	1.000	1.000	1.000	1.000	1.000			
DNN	0.893	0.996	0.802	0.797	0.902	0.781		χ2(3) = 87.45, p < 0.001	
XGB	0.993	0.999	0.985	0.989	0.994	0.985		χ2(2) = 5.67, p = 0.06	
RF	1.000	1.000	1.000	1.000	1.000	1.000			
Note:

AUC, area under receiver operating characteristic, SD, standard deviation, MCC, Matthew correlation coefficient, SVM, support vector machine, DNN, deep neural network, XGB, eXtreme gradient boosting, RF, random forest, the second is used to measure process time.

Figure 5 Model performance of test dataset—Leave-one-subject-out cross-validation.

AUC, area under receiver operating characteristic curve; MCC, Matthew correlation coefficient.

Under hold-out cross-validation, support vector machine still performed better than the remaining classifiers, with training samples, in terms of all metrics, including process time (see Table 11; Fig. 6). Deep neural network and eXtreme gradient boosting still did not perform as well as the remaining classifiers. The Stuart–Maxwell tests demonstrated significant results for both deep neural network (p < 0.001) and eXtreme gradient boosting (p = 0.018), indicating significant difference disagreement between these two algorithms and the observed data.

Figure 6 Model performance of test dataset—Holdout cross-validation.

AUC, area under receiver operating characteristic curve; MCC, Matthew correlation coefficient.

Table 11 Model performance: holdout cross-validation.

Sample	Method	Accuracy	AUC	MCC	Macro	Process time	Stuart–Maxwell test	
F1	Precision	Recall	
Train	SVM	1.000	1.000	1.000	1.000	1.000	1.000	0.23		
C5.0	0.950	0.996	0.903	0.933	0.948	0.920	0.59		
DNN	0.970	0.997	0.940	0.954	0.970	0.939	1.59		
XGB	0.886	0.974	0.775	0.809	0.869	0.770	0.75		
RF	0.978	1.000	0.957	0.980	0.989	0.972	0.39		
Test	SVM	1.000	1.000	1.000	1.000	1.000	1.000			
C5.0	1.000	1.000	1.000	1.000	1.000	1.000			
DNN	0.814	0.989	0.623	0.739	0.913	0.676	χ2(3) = 205.04, p < 0.001	
XGB	0.993	1.000	0.987	0.993	0.996	0.989	χ2(2) = 8.00, p = 0.018	
RF	1.000	1.000	1.000	1.000	1.000	1.000			
Note:

AUC, area under receiver operating characteristic; SD, standard deviation; MCC, Matthew correlation coefficient; SVM, support vector machine; DNN, deep neural network; XGB, eXtreme gradient boosting; RF, random forest; the second is used to measure process time.

Feature importance

In addition to making a comparison of the performance for the four prediction models, we also ranked the feature importance based on information gain and gain ratio (see Fig. 7). Information gain can be biased if features have a large number of possible outcomes, which may be corrected by gain ratio criteria (Kuhn & Johnson, 2013). From the perspective of information gain, or precisely how much a feature improves entropy (a measure of disorder), Hemoglobin A1c, age, triglyceride, low-density lipoprotein cholesterol, high-density lipoprotein cholesterol, and total cholesterol ranked as the top six most important features for predicting ICD-10-CM code. After correcting for possible bias, High-density lipoprotein cholesterol, Hemoglobin A1c, age, low-density lipoprotein cholesterol, triglyceride, and total cholesterol ranked as the top six important features. The greatest difference in the rankings, based upon information gain and gain ratio, is high-density lipoprotein cholesterol, ranked 5th by information gain, but ranked 1st by gain ratio. Further, BMI and glucose did not contribute anything to the class prediction of ICD-10-CM code for T2DM.

Figure 7 Importance of features.

Discussion

As mentioned at the beginning of our study, T2DM should be considered as a catastrophic threat to public health that is accompanied by huge financial and personal costs following the onset of T2DM. Therefore, obtaining the means of how to correctly diagnose T2DM patients in order to foster appropriate medical care for T2DM patients is inevitable and of great importance to the health-care profession. This study aimed to build an appropriate model for predicting ICD-10-CM code by utilizing bagging and boosting ensemble techniques for Asian T2DM patients. Our proposed model, based on support vector machine, performed well in terms of average accuracy, AUC, MCC, macro-averaging F1 score, macro-averaging precision, and macro-averaging recall. Based on information gain and gain ratio, our study also distinguished and ranked the top eight variables, including Hemoglobin A1c, age, triglyceride, low-density lipoprotein cholesterol, high-density lipoprotein cholesterol, and total Cholesterol, along with the habit of smoking, to predict ICD-10-CM codes for T2DM patients.

Although the performance metrics are not entirely consistent among T2DM-related studies that used machine-learning technique, it is still worthwhile to make a comparison between the current study and those studies with available performance metrics (see Table 12). Support vector machine was the best classifier in our study with accuracy, AUC, MCC, macro-averaging F1 score, macro-averaging precision, and macro-averaging recall metrics all equal to one. Prior studies utilized support vector machine also performed well but only in some metrics. For example, the study of Pei et al. (2019) achieved 0.908 accuracy rate for diabetes classification with non-invasive and easily gathered features. Talaei-Khoei & Wilson (2018) used machine-learning techniques to identify people at risk of developing T2DM and found the MCC metric of support vector machine was 0.922. Kagawa et al. (2017) combined expert knowledge and machine-learning approaches to determine whether a patient has T2DM. Among the five classifiers adopted, support vector machine achieved 0.909 in recall metric.

Table 12 Comparison of our study with state-of-the-art works.

Algorithms	Study	Accuracy	AUC	MCC	Precision	Recall	F1 score	
Support vector machine	This study	1	1	1	1	1	1	
Pei et al. (2019)	0.908	0.763	NA	0.903	0.908	0.905	
Talaei-Khoei & Wilson (2018)	NA	0.831	0.922	NA	0.683	NA	
Kagawa et al. (2017)	NA	NA	NA	0.8	0.909	NA	
Neural network	This study	0.788	0.986	0.566	0.910	0.620	0.684	
Talaei-Khoei & Wilson (2018)	NA	0.663	0.007	NA	0.41	NA	
Nilashi et al. (2017)	0.923	NA	NA	NA	NA	NA	
Esteban et al. (2017)	NA	NA	NA	0.930	0.960	0.940	
Random forest	This study	1	1	1	1	1	1	
Alghamdi et al. (2017)	0.840	NA	NA	0.844	0.994	0.913	
Note:

AUC, area under receiver operating characteristic; MCC, Matthew correlation coefficient; NA, not available.

Regarding studies that adopted neural network classifier, the study of Nilashi et al. (2017) achieved 0.923 accuracy rate while the study of Esteban et al. (2017) achieved 0.93 and 0.96 for precision and recall metrics, respectively. Finally, random forest also perfectly predicted ICD-10-CM code in our study for accuracy, AUC, MCC, macro-averaging F1 score, macro-averaging precision, and macro-averaging recall metrics. Alghamdi et al. (2017) adopted random forest to predict T2DM and achieved 0.844 and 0.994 for precision and recall, respectively. The reason that random forest performed quite well in our study may be due to the fact that random forest averages over multiple predictions to reduce the variance in the predictions (Provost & Fawcett, 2013).

Several interesting points can be derived from our findings as a whole. First, as suggested by prior literature (Lantz, 2015), our proposed predictive model implementing ensemble method (i.e., random forest and eXtreme gradient boosting) has performed, despite not being the best, satisfactorily with average accuracy, AUC, MCC, macro-averaging F1 score, macro-averaging precision, and macro-averaging recall being higher than 0.97 for all metrics among three resampling strategies. Future research may prove to implement these techniques that will lead to improved model predictive power.

Second, by using Asian samples, the findings determined in our study can be further compared with prior similar studies, and attention can be placed on the differences. For example, the link between obesity and T2DM remains uncertain (Eckel et al., 2011), BMI ranked the ninth important feature for predicting T2DM diagnosis in terms of both information gain and gain ratio. Future research can further explore why and how this difference comes to exist between eastern and western population samples.

Third, differing from most prior studies, our proposed models aimed to predict a multi-class classification task, which may provide more accurate predictions over and above binary classification tasking (Zhou, Tam & Fujita, 2016) since there may be numerous features that specifically identify a certain category. It is therefore of practical significance to apply a multi-class classification approach useful to predict ICD-10-CM code for T2DM patients.

Finally, our predictive model can be further developed into a clinical diagnosis support system, or even better when integrated into existing healthcare information systems aiming to support physicians, when diagnosing T2DM patients. By means of such a support system/function, physicians can better diagnose and foster medical care plans for T2DM patients to follow. The ability to predict disease sub-categories may assist and further remind physicians to early detect and manage possible complications in the earliest stages of disease onset.

One of the most important limitations found in our study is that we utilized only three ICD-10-CM codes pertinent to T2DM for predictive purposes. There are in fact many ICD-10-CM codes available for T2DM diagnosis and care; so, it is possible for future research to increase the number of ICD-10-CM codes in the predicted class in order to broaden diagnostic applications. In order to ensure as complete a data set as possible in building our model, we were required to remove those samples with missing data which resulted in only useable 149 samples extant. Future studies may choose to increase the sample size in order to enhance external the generalizability of the findings.

Conclusions

Our study adopted machine-learning techniques using 10 features adapted from Electronic Medical Records for identifying diagnosis code for T2DM patients. By adopting 10-fold, leave-one-subject-out, and holdout resampling strategy, support vector machine and random forest showed the best classification metrics in identifying an ICD-10-CM code for the test samples. These results demonstrated that our established model successfully achieved predictive and wholly appropriate ICD-10-CM code for T2DM patients to use. The implementation of our established predictive model in conjunction with using machine-learning algorithms along with data from Electronic Medical Records enables an in-depth exploration toward supporting diagnosis of T2DM patients. This approach may be easily applied within healthcare facilities which have implemented complete electronic medical record-keeping.

Supplemental Information

Supplemental Information 1 Dataset used for this study.

Click here for additional data file.

We want to show gratitude to Su-Ya Pan and Yao-Kun Cheng for carrying out the data application and collection used in this study.

Additional Information and Declarations

Competing Interests

Author Contributions

Ethics

Data Availability

The authors declare that they have no competing interests.

Kuang-Ming Kuo conceived and designed the experiments, performed the experiments, analyzed the data, prepared figures and/or tables, authored or reviewed drafts of the paper, and approved the final draft.

Paul Talley conceived and designed the experiments, performed the experiments, analyzed the data, prepared figures and/or tables, authored or reviewed drafts of the paper, and approved the final draft.

YuHsi Kao conceived and designed the experiments, performed the experiments, analyzed the data, prepared figures and/or tables, authored or reviewed drafts of the paper, and approved the final draft.

Chi Hsien Huang conceived and designed the experiments, performed the experiments, analyzed the data, prepared figures and/or tables, authored or reviewed drafts of the paper, and approved the final draft.

The following information was supplied relating to ethical approvals (i.e., approving body and any reference numbers):

The Institutional Review Board of E-Da Hospital approved the study (EMRP-107-048).

The following information was supplied regarding data availability:

Dataset used in this study is available as a Supplemental File.

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
