# Peer review of "A multi-class classification model for supporting the diagnosis of type II diabetes mellitus"

_PeerJ, doi:10.7717/peerj.9920_

## Round 0.1 · original submission · Major Revisions

Dear authors,

Two experts have reviewed your paper and they have found scientific merit in your work. However, there are some issues which you must address in a revised version of the text.

Best regards,
Dr Palazón-Bru (academic editor for PeerJ)

Reviewer 1 ·

Basic reporting

Clear and unambiguous, professional English used throughout?
The manuscript is well written and easy to read for the most part.

Literature references, sufficient field background/context provided?
Yes, the related work section is okay.

Structure conforms to PeerJ standards, discipline norm, or improved for clarity?
Yes, it meets the requirements.

Experimental design

Original primary research within Aims and Scope of the journal?
Can be improved.

Research question well defined, relevant & meaningful. It is stated how research fills an identified knowledge gap?
Yes.

Rigorous investigation performed to a high technical & ethical standard?
Can be improved.
Why this paper only compares tree based algorithms? As far as I know, there are many other advanced machine learning algorithms can be used to make predictions, such like neural networks, SVM, etc. Will bagging and boosting be better than these algorithms?

Methods described with sufficient detail & information to replicate?
Yes, methods can be replicated.

Validity of the findings

There are few explanations for the selected features, and there is little discussions how the features selected in this study different from other studies.

Additional comments

The authors built an appropriate model for predicting ICD-10-CM code by utilizing bagging and boosting ensemble techniques for Asian T2DM patients. This is a small innovations but it’s interesting.

·

Basic reporting

The paper is technically and logically sound and will contribute the scientific knowledge.
The introduction and literature,methodology, references are presented in systematic way.
The English proof reading is recommended for further improvement.
The figures are not suitable for publication and need proper modification.

Experimental design

The authors performed enough experiments for proposed method.

Validity of the findings

The experimental results are significants

Additional comments

The authors in this study design a multi-class classification model for supporting the diagnosis
of type II diabetes mellitus. To me the paper is well presented and suitable for publication in journal. Furthermore, 1 recommend to authors to incorporate the following changes for further improvement of the paper.
1. Remove all abbreviation from abstract and define in remaining parts of the paper such as F1, C5.0 etc.
2. Clearly mention your contribution in introduction section.
3. In last paragraph of introduction mention the paper organization
4. The literature need systematic way to write that present the existing methods issues and the important of proposed method.
5. This data set is linear or nonlinear.
6. The theoretical and mathematical back ground of the ML classifiers are necessary.

7. Authors only used machine learning algorithms for classification, I recommend to use Deep learning model for classification also and compared the prediction performance of both.
8. The authors also need to use LOSO, Hold out CV.
9. Used other performance evaluation metrics such, MCC, AUC ROC processing time of the model.
10. Why Random Forest algorithm performance high as compared to other models.
11. Statistically compare the ML models performance using McNamara’s test.
12. The figures and tables are not adjusted properly, all the figures X-axis and Y- axis titles are missing. Further the figures are not suitable for publication.
13. Compare you work with state of the art works
14. The conclusion is not clear
15. please use proper journal reference format all references are not properly used.

---

## Round 0.2 · Minor Revisions

Still pending some minor issues to be solved by the authors in a revised version of the text.

Reviewer 1 ·

Basic reporting

no comment

Experimental design

no comment

Validity of the findings

no comment

Additional comments

This manuscript has been greatly improved, but a couple of minor corrections should be incorporated:
1.Is it Matthew correlation coefficient or Mathew correlation coefficient?
2.The keywords should be modified. I think Bagging and Boosting can not represent the entire article.
3.Line 105-241: Can be simplified

---

## Round 0.3 · accepted · Accept

All the reviewers' concerns have been correctly addressed.

Reviewer 1 ·

Basic reporting

no comment

Experimental design

no comment

Validity of the findings

no comment

Additional comments

The authors have addressed all my concerns I had with the previous submission. I recommend an accept of this paper.